# Disentangled Information Quantification for Dataset Construction in Data-Centric AI

## Abstract

The rise of large-scale models has placed greater emphasis on Data-Centric AI (DCAI), which aims to build high-quality, scalable, and sustainable data assets. However, prevailing dataset construction relies on active learning with model-dependent criteria (e.g., uncertainty or confidence), typically anchored to a specific evaluation model. This coupling ties sample selection to architecture- and training-specific behavior, undermining transferability and reusability. We introduce Disentangled Generalizable Construction (DGC), a model-agnostic data selection method that, under a fixed sample budget, maximizes semantic coverage by selecting subsets that faithfully represent the datasets key semantic factors, thereby preserving utility across models and training settings. As a supporting module, the Orthogonal Factorization Soft-Quantization Autoencoder (OF-SQAE) yields stable, interpretable semantic factors; Invariant Attribute Information Quantification (IAIQ) quantifies per-sample informativeness in the latent space; and a coverage-driven greedy algorithm selects representative samples. Experiments on natural-image datasets and evaluations across diverse model architectures show strong results, indicating the generality and robustness of DGC.

## 1 Introduction

Data-Centric AI (DCAI) emphasizes that sustained improvements in model performance depend increasingly on the scale, quality, and structure of the data, rather than only architectural tweaks (Gohr et al., 2025; Achiam et al., 2023). At its core, DCAI shifts the focus of AI development from solely refining model architectures to systematically optimizing the entire data lifecycle — collection, cleaning, annotation, augmentation, and maintenance—with the aim of building high-quality, scalable, and sustainable data assets(Zha et al., 2025). These assets are intended to support diverse models over time, thereby improving overall performance, robustness, and adaptability. Within this perspective, dataset construction becomes a central step: under a fixed labeling budget, sample selection directly determines the coverage and informativeness of the supervision signal and is thus a prerequisite for reusable data assets(Mienye & Swart, 2024; Zha et al., 2023).

Existing work largely relies on active learning (AL) for sample selection(Tseng et al., 2025). The central idea is to prioritize labeling those unlabeled instances that are most valuable for the current model so as to maximize performance gains under a given annotation budget. In practice, uncertainty-based methods select samples where the predictive distribution has high entropy or small margin; diversity-based strategies use clustering or similarity metrics to cover representative regions of the input space; and training-dynamics approaches leverage gradient signals, loss trajectories, or influence functions to target samples with high impact on parameter updates. Despite their differences, these methods share a common premise: they evaluate sample value through a specific model and its current training state. This model dependence introduces bias, yielding unstable gains when transferring across architectures or tasks. Moreover, AL pipelines are sensitive to initial labeled sets, acquisition batch sizes, and model choices, which hurts stability and reproducibility(Huang et al., 2022). Crucially, under low selection rates (i.e., small acquisition budgets), such model-driven signals often struggle to balance efficiency and generalization. As a result, while AL can deliver strong one-off improvements for a particular model, it falls short of DCAI's goal of constructing long-term reusable data assets.

To address these issues, we propose a model-agnostic sample selection method, DGC (Disentangled Generalizable Construction). The objective is to maximize semantic coverage under a fixed acquisition budget: select a subset that best represents the dataset's key semantic factors. Concretely, we first learn an interpretable and stable disentangled semantic factor space from unlabeled data; we then introduce IAIQ (Invariant Attribute Information Quantification) to compute each sample's marginal contribution to semantic coverage in that space; finally, we perform a greedy search that iteratively admits samples in descending order of contribution until the budget is met, producing a high-information, low-redundancy subset. The entire procedure does not rely on downstream model predictions or training dynamics, and thus the selected subset transfers more reliably across different model architectures. To ensure a reliable factor space, we design OF-SQAE (Orthogonal Factorization Soft-Quantization Autoencoder), which incorporates orthogonal nonnegative matrix factorization and per-dimension soft quantization as structural inductive biases within an autoencoder, yielding stable, controllable, and interpretable latent representations that support robust IAIQ estimation.

In summary, our contributions are:

- We present DGC, a DCAI-oriented, model-agnostic sample selection method that uses semantic coverage-rather than model predictions-as the guiding criterion for construction high-value datasets.

- We introduce IAIQ, a computable metric of semantic coverage and per-sample marginal information in a disentangled factor space, paired with a greedy acquisition strategy for efficient selection.

- We design OF-SQAE to learn stable and interpretable disentangled representations, enabling robust, model-agnostic evaluation for selection.

- We validate the approach on synthetic and natural image datasets, demonstrating consistent advantages under low selection rates and across different model architectures.

## 2 RELATED WORK

### 2.1 DATA-CENTRIC AI

Data-Centric AI (DCAI) elevates data quality, coverage, and utility to a primary lever for ML performance, complementing model-centric advances (Miranda, 2021; Zha et al., 2023). The ecosystem spans data collection (Bhardwaj et al., 2014; Fernandez et al., 2018), annotation (Kutlu et al., 2020; Dong et al., 2023), preprocessing and curation (Hosseinzadeh et al., 2023; Krizhevsky et al., 2017), augmentation (Hendrycks et al., 2019; Zhang et al., 2020) and monitoring/iteration (Wongsuphasawat et al., 2015; Luo et al., 2018), framed as end-to-end data engineering pipelines (Polyzotis & Zaharia, 2021; Boecking et al., 2020; Zha et al., 2025). Active learning is a central mechanism for budgeted labeling, with uncertainty, diversity and gradient / sensitivity-based selection families (Lewis, 1995; Balcan et al., 2007; Nguyen & Smeulders, 2004; Sener & Savarese, 2017; Settles & Craven, 2008; Kirsch et al., 2019; Parvaneh et al., 2022; Tseng et al., 2025). However, many strategies depend on model-specific predictions or training dynamics and are sensitive to seeds and hyperparameters, which limits reuse across architectures and undermines reproducibility (Huang et al., 2022). We adopt a complementary, model-agnostic view that selects data by maximizing semantic coverage, aiming to construct durable, reusable assets for DCAI.

### 2.2 DISENTANGLED REPRESENTATION LEARNING

Unsupervised DRL seeks independent, semantically meaningful factors to improve interpretability and transfer (Locatello et al., 2019; Wang et al., 2024). The approaches span VAE (Higgins et al., 2017; Kim & Mnih, 2018; Kumar et al., 2017; Meo et al., 2024; Hsu et al., 2023), GAN (Chen et al., 2016; Lin et al., 2020; Jeon et al., 2021; Zhu et al., 2021), and diffusion-based models (Ren et al., 2021; Chen et al., 2023), usually using inductive biases or regularization to promote factor separation (Locatello et al., 2019). VAE variants often rely on KL/TC penalties to encourage independence, but this can trade off reconstruction fidelity and introduce stochasticity in the latents. We take a structural route: an autoencoding design that embeds orthogonal non-negative matrix factorization with per-dimension soft quantization, yielding stable, axis-aligned latents and controllable traversal

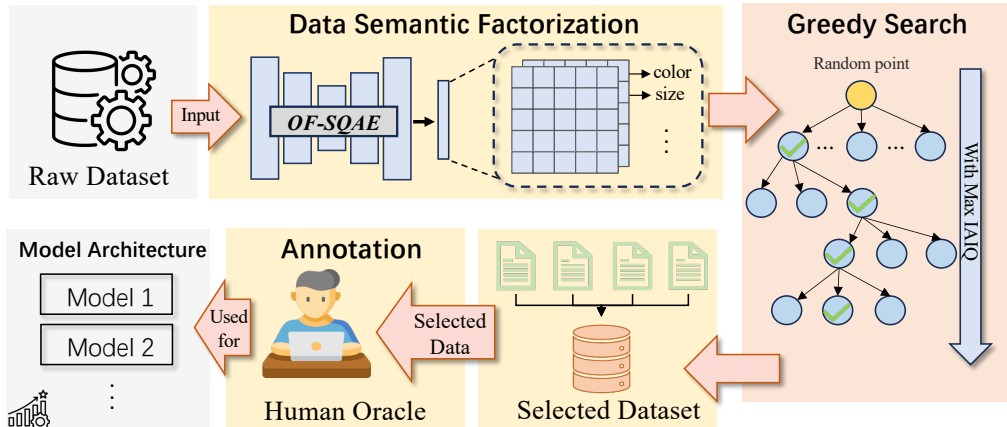

Figure 1: Overview of the proposed DGC method. The method integrates three main components: (1) the IAIQ theory, which quantifies the information content of unlabeled data in a model-agnostic manner; (2) the OF-SQAE model, which encodes data into a disentangled latent space to obtain structured and interpretable representations; and (3) a greedy search strategy, which iteratively selects a subset that maximizes the IAIQ score under a cardinality constraint. The resulting dataset demonstrates strong generalization, achieving consistently high performance across diverse model architectures.

without KL regularization—properties that are well-suited to downstream dataset evaluation and construction.

## 3 METHOD

This section details DGC (Fig. 1) in three parts: IAIQ (Sec. 3.1) provides a model-agnostic criterion for dataset information; OF-SQAE (Sec. 3.2) learns semantically disentangled representations; and a greedy selector (Sec. 3.3) maximizes IAIQ to construct a general-purpose subset.

### 3.1 INVARIANT ATTRIBUTE INFORMATION QUANTIFICATION

IAIQ is based on a fundamental principle in representation learning: a dataset with broader coverage of the underlying semantic space exhibits stronger generalization across models and tasks. Although neural architectures may differ, they ultimately aim to model the same latent factors. A subset that sufficiently captures these invariant attributes can effectively serve as a proxy for the full dataset. Therefore, quantifying how well a subset preserves such semantic diversity is essential for assessing its information content and its utility in general-purpose learning. IAIQ addresses this need through a model-agnostic metric that explicitly measures the semantic information richness of a data subset.

We formalize this by assuming that each input $x \in \mathcal{X}$ is generated from a latent semantic vector $\mathbf{s} \in \mathcal{S} \subseteq \mathbb{R}^D$, where each $s_d$ represents a semantically meaningful and invariant attribute (e.g., shape, color, orientation). We define an encoder function $f : \mathcal{X} \to \mathcal{S}$ and a decoder $g : \mathcal{S} \to \mathcal{X}$ that satisfy the self-consistency condition $g(f(x)) = x$. This bidirectional mapping enables semantic analysis in a disentangled latent space.

Based on this formalization, we propose the Invariant Attribute Information Quantification (IAIQ), a model-agnostic metric for evaluating the information content of a data subset $\mathcal{X}_S \subseteq \mathcal{X}$. The overall score is defined as the average per-attribute information:

$$\text{IAIQ}(\mathcal{X}_S) = \frac{1}{D} \sum_{d=1}^{D} I_d(\mathcal{X}_S) \quad [\text{bits}], \tag{1}$$

where $I_d(\mathcal{X}_S)$ quantifies the subset's information with respect to attribute $s_d$. This formulation captures both the diversity of values and the representational density of the subset, using base-2 logarithms to express results in bits.

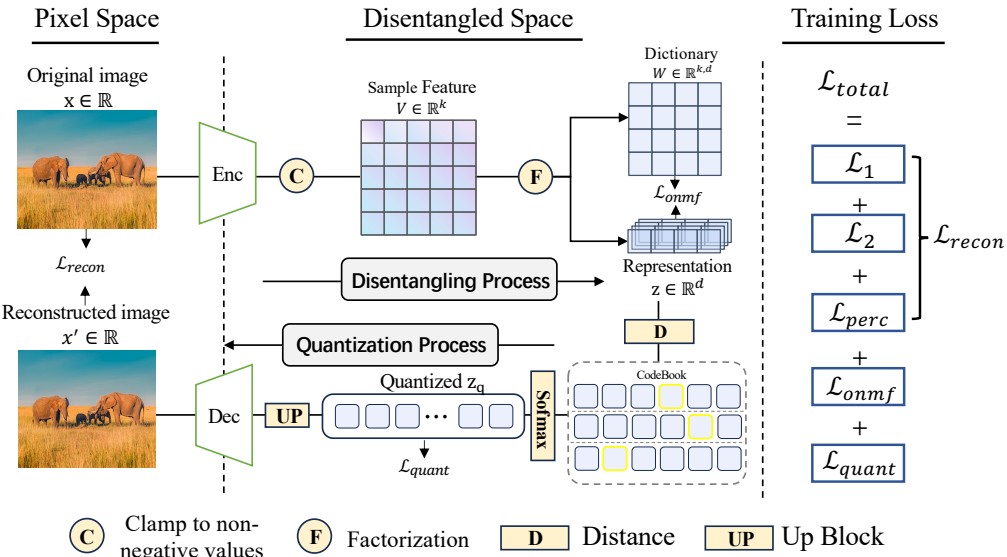

Figure 2: Overall architecture of OF-SQAE. The encoder maps images to a latent space, ONMF decomposes it into a dictionary $W$ and coordinates $z$, a per-dimension soft quantizer yields $z_q$, and the decoder reconstructs pixels. The loss combines $L_1$, $L_2$, $L_{\text{perc}}$, $L_{\text{onmf}}$, and $L_{\text{quant}}$.

Each term $I_d(\mathcal{X}_S)$ is factored into three interpretable components:

$$I_d(\mathcal{X}_S) = I_d^{\max} \cdot \gamma_d \cdot \rho, \tag{2}$$

With, $I_d^{\max}$ is the theoretical maximum information obtainable for attribute $s_d$ over the entire dataset, $\gamma_d$ measures the semantic coverage of the subset with respect to $s_d$, $\rho$ is a size-based correction factor that penalizes small subsets.

The maximal term $I_d^{\max}$ models an ideal upper bound on information for attribute $s_d$, defined as:

$$I_d^{\max} = H_d \cdot \Lambda, \quad \text{where} \quad \Lambda = \log_2(N+1), \tag{3}$$

with $N = |\mathcal{X}|$. Here, $H_d$ denotes the expected information per sample, computed as the Shannon entropy of the normalized latent distribution $p_d(s)$. We estimate $p_d(s)$ using kernel density estimation with an Epanechnikov kernel, which offers optimal efficiency with bounded support:

$$H_d = -\int_0^1 p_d(s) \log_2 p_d(s)\, ds. \tag{4}$$

A higher $H_d$ indicates more uniform semantic variability, and $\Lambda$ scales this to the dataset level.

To reflect how much of this maximal potential is captured by a subset, we define the attribute coverage coefficient and a size normalizer jointly:

$$\gamma_d = \frac{1}{m} \sum_{j=1}^m \frac{c_j}{k_{\text{eff},j}}, \qquad \rho = \frac{\log_2(n+1)}{\log_2(N+1)}, \tag{5}$$

where $k_{\text{eff},j}$ is the number of non-empty bins in the full dataset at scale $j$ and $c_j$ the count of those bins covered by $\mathcal{X}_S$; $n = |\mathcal{X}_S|$ and $N = |\mathcal{X}|$. The multi-scale design makes $\gamma_d \in [0, 1]$ robust to resolution choices and avoids overestimating sparse alignments, while $\rho$ smoothly penalizes small subsets and satisfies $\rho \to 1$ as $n \to N$.

In summary, IAIQ provides a principled, interpretable, and model-independent way to assess the semantic richness of data subsets. By integrating latent entropy, multi-scale support coverage, and size-aware normalization, it offers a robust proxy for evaluating the generalization potential of any selected dataset.

## 3.2 ORTHOGONAL FACTORIZATION SOFT-QUANTIZATION AUTOENCODER

We design OF-SQAE to extract structured, interpretable, and axis-aligned latent factors. As illustrated in Fig. 2, the model is an autoencoder with encoder $f : \mathcal{X} \to \mathbb{R}^k$ and decoder $g : \mathbb{R}^d \to \mathcal{X}$. The encoder maps an input $x$ to features $v = f(x) \in \mathbb{R}^k$, which are decomposed into a disentangled code $z \in \mathbb{R}^d$ via a non-negative matrix factorization with orthogonality, $z \approx vW^\top$, where $W \in \mathbb{R}_+^{d \times k}$ and $z \in \mathbb{R}_+^d$ while $WW^\top \approx I_d$. Crucially, imposing non-negativity on $(W, z)$ and near-orthogonality on the rows of $W$ collapses many equivalent NMF factorizations (up to permutation/scale) into a much smaller solution set and anchors each row of $W$ to a distinct, non-overlapping semantic direction; together with an $\ell_1$ sparsity on $z$, this shrinkage of the solution space biases the representation toward axis-aligned, interpretable factors. The quantized representation is finally decoded as $\hat{x} = g(z)$.

Training balances reconstruction fidelity, factor identifiability, and stable discretization:

$$\mathcal{L}_{\text{total}} = \mathcal{L}_{\text{L1}} + \mathcal{L}_{\text{L2}} + \lambda_1 \mathcal{L}_{\text{perc}} + \lambda_2 \mathcal{L}_{\text{onmf}} + \lambda_3 \mathcal{L}_{\text{quant}}. \tag{6}$$

We instantiate the ONMF penalty as

$$\mathcal{L}_{\text{onmf}} = \log\big(1 + \|\mathbf{V} - \mathbf{W}\mathbf{Z}\|_2^2\big) + \|\mathbf{Z}\|_1 + \log\big(1 + \|\mathbf{W}\mathbf{W}^\top - \mathbf{I}_d\|_F\big), \tag{7}$$

where $\mathbf{V}$ are encoder features, $\mathbf{Z}$ the non-negative codes, and $\mathbf{W}$ a non-negative, approximately orthogonal dictionary.

To further regularize the geometry of $z$ and make each coordinate traversable, we apply a per-dimension *soft* quantization that maps $z_i$ to a convex combination of a learnable 1D codebook $\mathbf{c}_i \in \mathbb{R}^{K_i}$:

$$\mathbf{a}_i = \text{softmax}\left(-\frac{|\mathbf{z}_i \mathbf{1}_{K_i}^\top - \mathbf{c}_i|}{\tau}\right), \quad z_{q,i} = \mathbf{a}_i^\top \mathbf{c}_i, \quad \mathcal{L}_{\text{quant}} = \|\mathbf{z}_q - \mathbf{z}\|_2^2. \tag{8}$$

This discretization acts as an explicit inductive bias toward separable, symbol-like coordinates: it suppresses spurious micro-variations, yields stable axis-wise semantics, and enables dimension-wise traversal by stepping along a single code index—*without* requiring a variational KL term.

A formal convergence analysis of the ONMF optimization (based on projected gradient updates under mild smoothness and boundedness assumptions) is provided in Appendix A.1

## 3.3 GENERAL-PURPOSE DATASET CONSTRUCTION

The final component of our method uses the OF-SQAE model and the IAIQ theory to construct a compact, high-information dataset tailored for robust classification models. We operationalize the IAIQ metric by using the quantized latent codes $\mathbf{z}_q$ from OF-SQAE as the concrete semantic attributes $\mathbf{s}$. The guiding principle is to build a subset that maximizes semantic coverage. The formal objective remains finding a subset $\mathcal{X}_S$ of size $n$ that maximizes the global IAIQ score:

$$\mathcal{X}_S^* = \text{argmax}_{\mathcal{X}_S \subseteq \mathcal{X}, |\mathcal{X}_S| = n} \text{IAIQ}(\mathcal{X}_S). \tag{9}$$

As this is computationally intractable, we employ a greedy forward selection algorithm. However, instead of re-evaluating the expensive global IAIQ score at each step, our algorithm selects the sample with the highest marginal contribution score. This score is designed to directly reward samples that populate under-represented regions of the latent space across all dimensions. Specifically, the contribution of a candidate sample is calculated based on the inverse of the occupancy counts of the bins it falls into. This ensures that a sample landing in a sparsely populated bin receives a higher score. Algorithmic details are provided in Appendix A.2. The resulting dataset, $\mathcal{X}_S$, is by construction a compact and semantically diverse subset, specifically optimized for training high-performance classification models.

## 4 EXPERIMENTS

This section systematically evaluates our approach. Section 4.1 presents the experimental setup. We then assess disentanglement in Section 4.2, validate the IAIQ theory in Section 4.3, and evaluate cross-model generalization to demonstrate the effectiveness of our dataset construction in Section 4.4.

## 4.1 SETUP

**Backbones and datasets.** We evaluate three aspects of our approach: (i) disentanglement on Shapes3D Kim & Mnih (2018) and Cars3D Reed et al. (2015); (ii) IAIQ validation on Shapes3D Kim & Mnih (2018); and (iii) dataset construction on STL-10 Coates et al. (2011), CIFAR-10 Krizhevsky et al. (2009), and Imagenette[1]. For generalization tests of constructed subsets, we train multiple classification backbones (ResNet-18 He et al. (2016), SENet-18 Hu et al. (2018), VGG-11 Simonyan & Zisserman (2014), Inception Szegedy et al. (2016), and MobileNet Sandler et al. (2018)). A full description of each dataset and splits appears in Appendix A.3.

**Evaluation.** For disentanglement, we report DCI Eastwood & Williams (2018) and the FactorVAE score Kim & Mnih (2018) using `disentanglement_lib` Locatello et al. (2019). For IAIQ validation (multitask and single task on Shapes3D), we measure top-1 accuracy per factor and the mean across factors. For dataset construction, we assess subset generalization by training multiple backbones on the same selected data and reporting mean $\pm$ std test accuracy across models. Precise metric definitions and evaluation protocols are provided in Appendix A.4.

**Baselines.** *Disentanglement:* FactorVAE Kim & Mnih (2018), $\beta$-TCVAE Chen et al. (2018), CD-VAE Huang et al. (2025), DAVA Estermann & Wattenhofer (2023), DisCo Ren et al. (2021), LatentDisco Voynov & Babenko (2020), GANSpace Härkönen et al. (2020), InfoGAN-CR Lin et al. (2020), CL-Dis Jin et al. (2024), DisDiff Yang et al. (2023). *Dataset construction (active learning):* BADGE Ash et al. (2019), ALMix Parvaneh et al. (2022), SAAL Kim et al. (2023), LearningLoss Yoo & Kweon (2019) and CoreSet Sener & Savarese (2017). All baselines follow authors' official code/hyperparameters when available; otherwise we adopt settings from the original papers.

**Training details.** All experiments run on NVIDIA L20 GPUs under Ubuntu 22.04 with CUDA 12.4, PyTorch 2.4.1, and Python 3.8. OF-SQAE is trained end-to-end to obtain disentangled, softly quantized latents; DGC then selects high-information subsets via the greedy procedure in Section 3.3. For active-learning baselines, we start from a 5% labeled seed and conduct 7 rounds with an additional +5% labels per round under identical augmentation and optimization protocols. Backbone training and the MTANNet (Liu et al., 2019) setup for IAIQ validation follow standard practice; complete hyperparameters, schedules, and data augmentations are summarized in Appendix A.5.

Table 1: Disentanglement results on synthetic datasets.

| Method | Architecture | | Shapes3D | | Cars3D | |
|---|---|---|---|---|---|---|
| | Type | Optimization | FactorVAE ↑ | DCI ↑ | FactorVAE ↑ | DCI ↑ |
| FactorVAE | VAE | End-to-end | $0.840 \pm 0.066$ | $0.611 \pm 0.082$ | $0.906 \pm 0.052$ | $0.161 \pm 0.019$ |
| $\beta$-TCVAE | VAE | End-to-end | $0.873 \pm 0.074$ | $0.613 \pm 0.114$ | $0.855 \pm 0.082$ | $0.140 \pm 0.019$ |
| CD-VAE | VAE | Alternating | $0.970 \pm 0.040$ | $0.630 \pm 0.040$ | $0.830 \pm 0.100$ | $0.260 \pm 0.030$ |
| DAVA | VAE | End-to-end | $0.820 \pm 0.030$ | $0.780 \pm 0.030$ | $0.940 \pm 0.100$ | $0.230 \pm 0.040$ |
| InfoGAN-CR | GAN | End-to-end | $0.587 \pm 0.058$ | $0.478 \pm 0.055$ | $0.411 \pm 0.013$ | $0.020 \pm 0.011$ |
| LatentDisco | GAN | Alternating | $0.805 \pm 0.064$ | $0.380 \pm 0.062$ | $0.852 \pm 0.039$ | $0.216 \pm 0.072$ |
| GANSpace | GAN | Alternating | $0.788 \pm 0.091$ | $0.284 \pm 0.034$ | $0.932 \pm 0.018$ | $0.209 \pm 0.031$ |
| DisCo | GAN | Alternating | $0.877 \pm 0.031$ | $0.708 \pm 0.048$ | $0.855 \pm 0.074$ | $0.271 \pm 0.037$ |
| DisDiff | Diffusion | Alternating | $0.902 \pm 0.043$ | $0.723 \pm 0.013$ | $\mathbf{0.976 \pm 0.018}$ | $0.232 \pm 0.019$ |
| CL-Dis | Diffusion | Alternating | $0.952 \pm 0.017$ | $0.731 \pm 0.045$ | *N/A* | *N/A* |
| **OF-SQAE (Ours)** | AE | End-to-end | $\mathbf{0.988 \pm 0.010}$ | $\mathbf{0.903 \pm 0.015}$ | $0.940 \pm 0.000$ | $\mathbf{0.336 \pm 0.030}$ |

## 4.2 DISENTANGLEMENT EVALUATION

We evaluate OF-SQAE on Shapes3D and Cars3D using the DCI and FactorVAE metrics (Table 1). In Shapes3D, where the ground truth factors are balanced and mostly independent, OF- SQAE achieves the best scores, with DCI $0.903 \pm 0.015$ and FactorVAE $0.988 \pm 0.010$. This matches or exceeds the strongest multi-stage pipelines while retaining single-stage efficiency. On Cars3D, which exhibits substantial intra-class variation and viewpoint changes, OF-SQAE achieves the highest DCI

---

[1]Jeremy Howard, Imagenette: A smaller subset of 10 easily classified classes from ImageNet, fast.ai, https://github.com/fastai/imagenette.

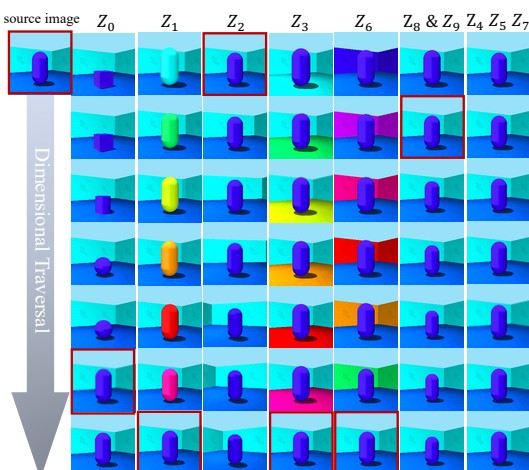

Figure 3: Dimension traversal interpolates latent dimensions with the rest fixed.

Table 2: Ablation study of OF-SQAE on Shapes3D.

| Components | | Metrics | |
|---|---|---|---|
| ONMF | Quantization | DCI ↑ | FactorVAE ↑ |
| ✗ | ✗ | 0.3486 | 0.4718 |
| ✓ | ✗ | 0.7007 | 0.9568 |
| ✓ | ✓ | 0.9028 | 0.9884 |

($0.336 \pm 0.030$) and a competitive FactorVAE ($0.940$), close to the best diffusion-based baseline, indicating robustness under stronger factor coupling.

Qualitative traversal on Shapes3D (Fig. 3) further confirms axis–semantics alignment. Starting from a source image, we vary one latent coordinate $z_k$ across its codebook while fixing the others; the decoded images show coherent control over individual factors (e.g., shape, object hue, azimuth, scale) with minimal leakage to unrelated attributes. The samples remain clean and consistent, suggesting well-regularized, interpretable latents.

An ablation on Shapes3D (Table 2) isolates the contributions of each component. Training with reconstruction losses only yields DCI $0.3486$ and FactorVAE $0.4718$. Adding the ONMF loss raises performance to DCI $0.7007$ and FactorVAE $0.9568$, showing that ONMF is the primary driver of factor separation. Appending the soft quantization head further refines the codebooks and reduces

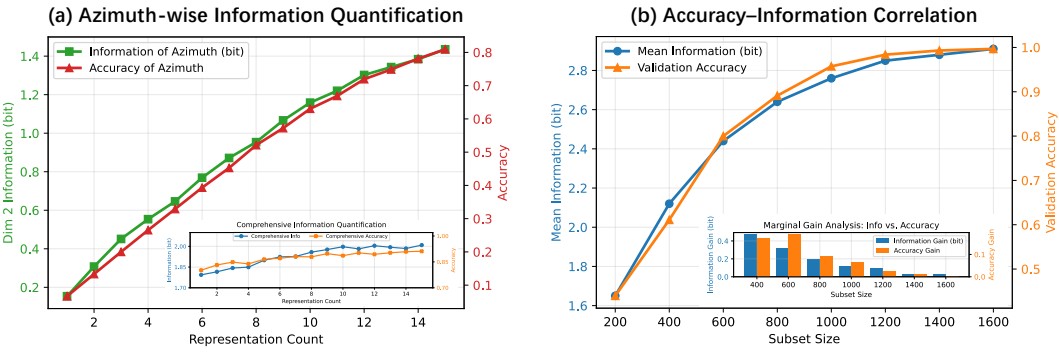

Figure 4: IAIQ validation on Shapes3D. (a) Single-attribute (azimuth) quantification: IAIQ (bits) tracks single-task accuracy as representation count grows. (b) Multi-attribute mean IAIQ vs. multi-task accuracy across subset sizes, showing tight correlation and diminishing returns.

Table 3: Generalization of selected subsets across multiple backbones.

| Dataset | Method | Predefined Selection Rate | | | | | | | |
|---|---|---|---|---|---|---|---|---|---|
| | | 5% | 10% | 15% | 20% | 25% | 30% | 35% | 40% |
| CIFAR-10 | SAAL | | $0.5439_{\pm 0.1609}$ | $0.6790_{\pm 0.1096}$ | $0.7207_{\pm 0.0990}$ | $0.7669_{\pm 0.0716}$ | $0.7900_{\pm 0.0694}$ | $0.8103_{\pm 0.0558}$ | $0.8125_{\pm 0.0590}$ |
| | BADGE | | $0.5181_{\pm 0.1902}$ | $0.6712_{\pm 0.1192}$ | $0.7062_{\pm 0.0962}$ | $0.7693_{\pm 0.0758}$ | $0.7829_{\pm 0.0772}$ | $0.8010_{\pm 0.0676}$ | $0.8082_{\pm 0.0636}$ |
| | ALMix | $0.4460_{\pm 0.1565}$ | $0.5279_{\pm 0.1734}$ | $0.7021_{\pm 0.0743}$ | $0.7406_{\pm 0.0603}$ | $0.7707_{\pm 0.0583}$ | $0.7950_{\pm 0.0491}$ | $0.8158_{\pm 0.0405}$ | $0.8185_{\pm 0.0396}$ |
| | CoreSet | | $0.5354_{\pm 0.1769}$ | $0.6795_{\pm 0.1081}$ | $0.7168_{\pm 0.0889}$ | $0.7722_{\pm 0.0556}$ | $0.7957_{\pm 0.0521}$ | $0.8156_{\pm 0.0450}$ | $0.8177_{\pm 0.0456}$ |
| | LearningLoss | | $0.4301_{\pm 0.1564}$ | $0.5044_{\pm 0.1485}$ | $0.6072_{\pm 0.1322}$ | $0.6789_{\pm 0.1308}$ | $0.6956_{\pm 0.1447}$ | $0.7286_{\pm 0.1227}$ | $0.7369_{\pm 0.1079}$ |
| | **DGC (Ours)** | $\mathbf{0.5290}_{\pm 0.1358}$ | $\mathbf{0.6573}_{\pm 0.1079}$ | $\mathbf{0.7310}_{\pm 0.0646}$ | $\mathbf{0.7615}_{\pm 0.0532}$ | $\mathbf{0.7934}_{\pm 0.0448}$ | $\mathbf{0.8089}_{\pm 0.0424}$ | $\mathbf{0.8227}_{\pm 0.0374}$ | $\mathbf{0.8278}_{\pm 0.0370}$ |
| STL-10 | SAAL | | $0.4638_{\pm 0.1034}$ | $0.5196_{\pm 0.1137}$ | $0.5653_{\pm 0.1069}$ | $0.5838_{\pm 0.1032}$ | $0.6277_{\pm 0.0899}$ | $0.6550_{\pm 0.0735}$ | $0.6661_{\pm 0.0772}$ |
| | BADGE | | $0.4654_{\pm 0.1289}$ | $0.5344_{\pm 0.1029}$ | $0.5420_{\pm 0.1279}$ | $0.5893_{\pm 0.1176}$ | $0.6280_{\pm 0.0929}$ | $0.6205_{\pm 0.1326}$ | $0.6564_{\pm 0.0736}$ |
| | ALMix | $0.3629_{\pm 0.1344}$ | $0.4786_{\pm 0.0877}$ | $0.5491_{\pm 0.0929}$ | $0.6003_{\pm 0.0734}$ | $0.6255_{\pm 0.0761}$ | $0.6406_{\pm 0.0908}$ | $0.6746_{\pm 0.0473}$ | $0.6774_{\pm 0.0479}$ |
| | CoreSet | | $0.4606_{\pm 0.0938}$ | $0.5422_{\pm 0.0602}$ | $0.5883_{\pm 0.0529}$ | $0.5874_{\pm 0.1100}$ | $0.6358_{\pm 0.0733}$ | $0.6692_{\pm 0.0508}$ | $0.6737_{\pm 0.0475}$ |
| | LearningLoss | | $0.3832_{\pm 0.1079}$ | $0.4253_{\pm 0.1086}$ | $0.4925_{\pm 0.1172}$ | $0.5166_{\pm 0.1105}$ | $0.5362_{\pm 0.1251}$ | $0.5758_{\pm 0.1080}$ | $0.5610_{\pm 0.1028}$ |
| | **DGC (Ours)** | $\mathbf{0.4384}_{\pm 0.0911}$ | $\mathbf{0.5070}_{\pm 0.0884}$ | $\mathbf{0.5848}_{\pm 0.0616}$ | $\mathbf{0.6267}_{\pm 0.0531}$ | $\mathbf{0.6531}_{\pm 0.0523}$ | $\mathbf{0.6805}_{\pm 0.0454}$ | $\mathbf{0.6938}_{\pm 0.0470}$ | $\mathbf{0.7077}_{\pm 0.0484}$ |
| Imagenette | SAAL | | $0.5278_{\pm 0.2295}$ | $0.6065_{\pm 0.2122}$ | $0.6653_{\pm 0.1712}$ | $0.6696_{\pm 0.2289}$ | $0.7053_{\pm 0.2122}$ | $0.7126_{\pm 0.2163}$ | $0.7360_{\pm 0.1756}$ |
| | BADGE | | $0.5698_{\pm 0.1540}$ | $0.6556_{\pm 0.1251}$ | $0.6553_{\pm 0.1933}$ | $0.6973_{\pm 0.1938}$ | $0.6541_{\pm 0.2868}$ | $0.6779_{\pm 0.2577}$ | $0.7278_{\pm 0.1863}$ |
| | ALMix | $0.4657_{\pm 0.1737}$ | $0.5360_{\pm 0.2256}$ | $0.6171_{\pm 0.2286}$ | $0.6493_{\pm 0.2250}$ | $0.6739_{\pm 0.2392}$ | $0.7143_{\pm 0.2213}$ | $0.7385_{\pm 0.2045}$ | $0.7256_{\pm 0.2443}$ |
| | CoreSet | | $0.5561_{\pm 0.1816}$ | $0.6580_{\pm 0.1258}$ | $0.7122_{\pm 0.0853}$ | $0.7063_{\pm 0.1441}$ | $0.7617_{\pm 0.0871}$ | $0.7831_{\pm 0.1016}$ | $0.7941_{\pm 0.0689}$ |
| | LearningLoss | | $0.4885_{\pm 0.1375}$ | $0.5375_{\pm 0.1640}$ | $0.5929_{\pm 0.1762}$ | $0.5977_{\pm 0.2600}$ | $0.6501_{\pm 0.2208}$ | $0.6818_{\pm 0.2011}$ | $0.6787_{\pm 0.2103}$ |
| | **DGC (Ours)** | $\mathbf{0.5652}_{\pm 0.0638}$ | $\mathbf{0.6512}_{\pm 0.0939}$ | $\mathbf{0.7218}_{\pm 0.0660}$ | $\mathbf{0.7667}_{\pm 0.0477}$ | $\mathbf{0.7827}_{\pm 0.0559}$ | $\mathbf{0.8074}_{\pm 0.0385}$ | $\mathbf{0.8150}_{\pm 0.0514}$ | $\mathbf{0.8331}_{\pm 0.0338}$ |

All methods start from the same randomly selected 5% labeled seed. Numbers are mean$_{\pm \text{std}}$ across backbones.

residual cross-factor mixing, reaching DCI 0.9028 and FactorVAE 0.9884. Overall, OF-SQAE delivers strong disentanglement in a single stage and provides practical, interpretable latents for downstream tasks such as dataset evaluation and construction.

### 4.3 VALIDATION OF THE IAIQ THEORY

We validate IAIQ on Shapes3D, which permits controlled variation of semantic factors. Latents are extracted by OF-SQAE (Sec. 4.2), and all runs use the same MTANet evaluator and training protocol for comparability. In Fig. 4(a), we fix the subset size at 800 and vary the number of azimuth categories from 1 to 15 while leaving other factors unconstrained; the IAIQ assigned to azimuth and the corresponding single-task accuracy co-vary—both increase and then saturate—and the inset shows that averaging IAIQ over all factors similarly tracks multi-task accuracy. In Fig. 4(b), we sample subsets of size 200–1600; mean IAIQ and overall validation accuracy increase with subset size before plateauing, with the inset highlighting diminishing marginal gains. Taken together, the single-factor and multi-factor experiments provide empirical support for the analysis in Sec. 3.1: model accuracy co-trends with IAIQ, and broadening the coverage of underlying representation types (i.e., increasing factor diversity or sample budget) consistently raises both IAIQ and downstream accuracy until saturation. Thus, IAIQ serves as a reliable, model-agnostic surrogate for dataset informativeness and a practical criterion for data selection.

### 4.4 COMPARISON OF DATASET GENERALIZATION

We evaluate DGC on CIFAR-10, STL-10 and Imagenette with labeled budgets from 5% to 40%. All methods (SAAL, BADGE, ALMix, CoreSet, Learning Loss) construct training subsets using ResNet-18. To evaluate cross-architecture generalization, we then train SENet-18, VGG-11, Inception, and MobileNet on the distinct subsets selected by each method at each budget, and report mean$_{\pm \text{std}}$ accuracy aggregated across backbones.

As summarized in Table 3, DGC achieves the best average accuracy across most budgets and datasets, with the largest margins in the low-budget regime. On CIFAR-10, it leads by ∼11 points at 10% (0.657 compared with 0.544), by 2–3 points at 15–25%, and remains ahead at 30–40% (e.g., 0.828 compared with 0.819 at 40%). On STL-10, gains are consistent from 10% to 40% (+2.8 at 10%, +4.3 at 15%, +3.0 at 40%). On Imagenette, improvements are pronounced across all rates (+8.1 at 10%, +6.4 at 15%, +3.9 at 40%). DGC also reduces cross-backbone variance: the standard deviation is typically lower on CIFAR-10/STL-10, and on Imagenette it is ≈0.04–0.09, indicating more reliable transfer to unseen architectures.

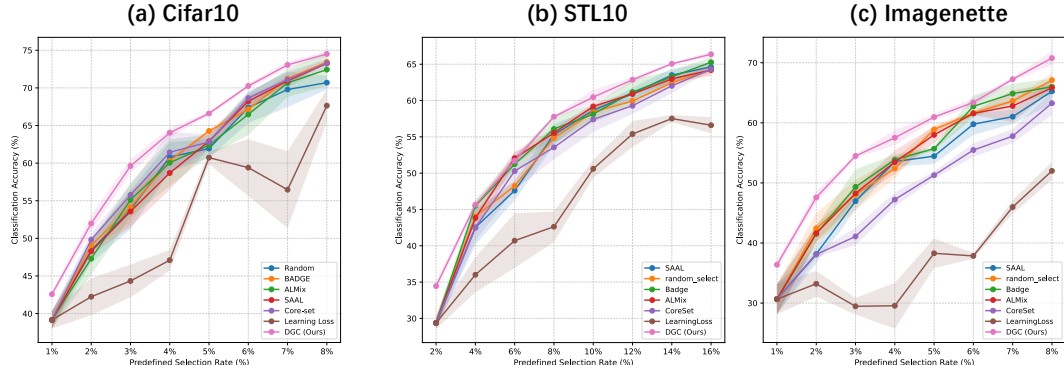

Figure 5: Subset selection under low label budgets (1% per round, up to 8%) on CIFAR-10, STL-10, and Imagenette. Curves report mean accuracy across backbones; shaded areas show standard deviation.

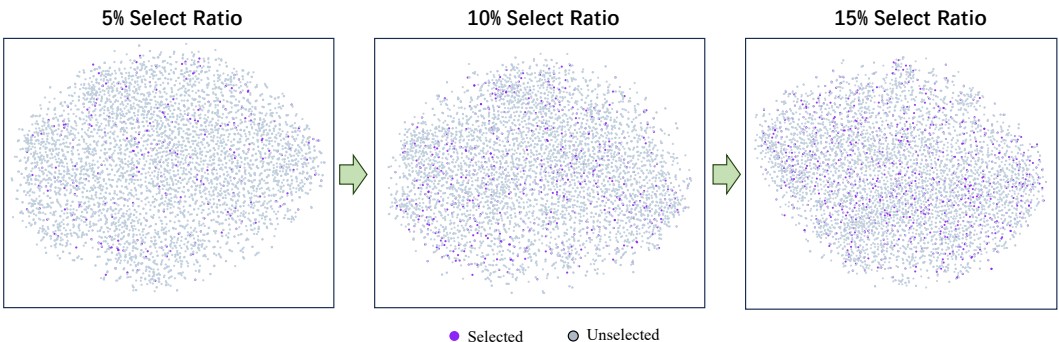

Figure 6: t-SNE visualization of OF-SQAE latents on CIFAR-10. Selected and unselected samples are encoded and projected to 2D; panels correspond to selection rates of 5%, 10%, and 15%. Selected points cover diverse regions of the feature space.

We further study a progressive low-budget setting starting from 1% labels and adding 1% per round for seven rounds (construction and evaluation both with ResNet-18 to isolate selector behavior). Figure 5 shows that DGC attains consistently higher accuracy at every round, exhibits narrower run-to-run variability, and reaches its plateau with fewer labels. These observations, together with the cross-backbone results in Table 3, indicate that DGC's coverage-driven criterion selects subsets that (i) generalize across architectures and (ii) are especially beneficial when labels are scarce, yielding rapid gains early and diminishing returns as the budget increases.

## 5 CONCLUSION

We present DGC, a model-agnostic dataset construction framework that selects high-utility subsets by maximizing semantic coverage in a disentangled latent space. OF-SQAE combines orthogonal NMF with per-dimension soft quantization to produce stable, interpretable latents, and IAIQ quantifies attribute-wise information to guide a simple greedy selector. On synthetic data, OF-SQAE shows strong disentanglement and IAIQ aligns with task performance; on CIFAR-10, STL-10, and Imagenette, DGC consistently outperforms baselines across architectures and label budgets, with clear gains at low labels. These results indicate that prioritizing semantic coverage yields compact, transferable datasets and supports practical pipelines for data-centric AI.

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

# A APPENDIX

## A.1 CONVERGENCE ANALYSIS OF ONMF

**Setup.** Recall the ONMF penalty in Eq. equation 7 and define

$$\mathcal{L}_{\text{onmf}}(\mathbf{W}, \mathbf{Z}) = \phi(\mathbf{W}, \mathbf{Z}) + \|\mathbf{Z}\|_1 + \psi(\mathbf{W}), \quad \phi(\mathbf{W}, \mathbf{Z}) = \log\big(1 + \|\mathbf{V} - \mathbf{W}\mathbf{Z}\|_2^2\big), \ \psi(\mathbf{W}) = \log\big(1 + \|\mathbf{W}\mathbf{W}^\top - \mathbf{I}_d\|_F\big),$$

with non-negativity constraints $\mathbf{W} \geq 0$, $\mathbf{Z} \geq 0$. We consider projected gradient updates

$$\mathbf{W}^{t+1} = \Pi_{\geq 0}\big(\mathbf{W}^t - \eta_t \nabla_{\mathbf{W}} \mathcal{L}_{\text{onmf}}(\mathbf{W}^t, \mathbf{Z}^t)\big), \qquad \mathbf{Z}^{t+1} = \Pi_{\geq 0}\big(\mathbf{Z}^t - \eta_t \nabla_{\mathbf{Z}} \mathcal{L}_{\text{onmf}}(\mathbf{W}^t, \mathbf{Z}^t)\big),$$

where $\Pi_{\geq 0}$ denotes projection onto the non-negative orthant and $\eta_t > 0$ is the step size.

**Assumptions.** (i) The sequence $\{(\mathbf{W}^t, \mathbf{Z}^t)\}$ is bounded (standard in practice via initialization, weight decay, or column-wise normalization of $\mathbf{W}$); (ii) $\mathcal{L}_{\text{onmf}}$ has Lipschitz-continuous gradients on the bounded domain induced by (i); (iii) Step sizes are chosen as $0 < \eta_t \leq \bar{\eta} < \frac{1}{L}$, where $L$ is a (local) Lipschitz constant of $\nabla \mathcal{L}_{\text{onmf}}$ on the working set.

**Proposition (Descent and criticality).** Under (i)–(iii), the projected gradient iterates satisfy:

(a) *Sufficient decrease:* there exists $c > 0$ such that

$$\mathcal{L}_{\text{onmf}}(\mathbf{W}^{t+1}, \mathbf{Z}^{t+1}) \leq \mathcal{L}_{\text{onmf}}(\mathbf{W}^t, \mathbf{Z}^t) - c\left\|G^t\right\|_F^2,$$

where $G^t$ denotes the concatenated projected gradient at $(\mathbf{W}^t, \mathbf{Z}^t)$.

(b) *Convergence to stationary points:* $\lim_{t \to \infty} \|G^t\|_F = 0$. Every accumulation point of $\{(\mathbf{W}^t, \mathbf{Z}^t)\}$ is a first-order stationary point (critical point) of the constrained problem.

**Proof sketch.** By (ii), the descent lemma applies to the smooth part $\phi(\mathbf{W}, \mathbf{Z})$ and $\psi(\mathbf{W})$. The non-smooth component $\|\mathbf{Z}\|_1$ is handled via the projected (equivalently, proximal with $\ell_1$ and non-negativity) step, which guarantees a decrease for step sizes $\eta_t < 1/L$. The projection $\Pi_{\geq 0}$ is non-expansive, so the composite step admits a standard sufficient-decrease bound (e.g., via the three-point inequality). Summing the decrease over $t$ shows $\sum_t \|G^t\|_F^2 < \infty$, hence $\|G^t\|_F \to 0$, establishing (b). Because $\phi$ and $\psi$ are real-analytic and the feasible set is semi-algebraic, the objective satisfies the Kurdyka–Łojasiewicz (KŁ) property on bounded sets; thus, the whole sequence converges to a critical point (rather than merely having critical accumulation points).

Figure 7 illustrates the above behavior in practice. Figure (a) plots the average inter-vector correlation of the rows of $\mathbf{W}$ during training, which rapidly drops toward 0, indicating near-orthogonality as enforced by $\psi(\mathbf{W})$. Figure (b) shows the learned inter-vector correlation matrix: strong self-correlation on the diagonal and minimal off-diagonal correlation, consistent with the stationary-point characterization of the projected gradient dynamics.

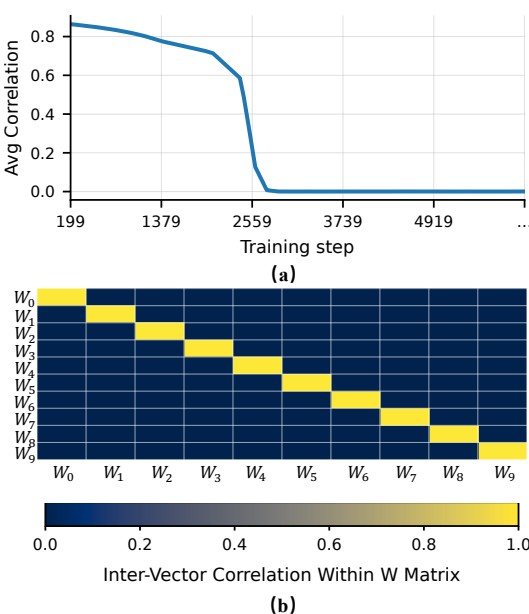

Figure 7: ONMF convergence on SHAPES3D. (a) Average inter-vector correlation of $\mathbf{W}$ versus training steps, approaching 0; (b) correlation matrix of $\mathbf{W}$ at convergence, showing a nearly diagonal structure.

### A.2 PSEUDOCODE OF THE DGC ALGORITHM

Algorithm 1 builds a labeled subset $\mathcal{X}_S$ of size $n$ from an unlabeled pool $\mathcal{X}$ using disentangled, softly quantized latents $\mathbf{Z} \in \mathbb{R}^{N \times D}$ produced by OF-SQAE. At each step, DGC greedily selects the sample with the largest *marginal contribution* to semantic coverage, measured via multi-scale binning along each latent dimension $d \in \{1, \ldots, D\}$ and scale $k \in \mathcal{K}$. The per-sample score averages the inverse occupancy of the bins it falls into across dimensions and scales, serving as an efficient surrogate for the IAIQ objective - higher scores indicate a higher expected gain in semantic information.

### A.3 DESCRIPTION OF THE DATASET

Our experiments are conducted on several benchmark datasets, categorized as either synthetic for controlled analysis or natural images for real-world performance evaluation.

#### A.3.1 SYNTHETIC DATASETS

These datasets feature controlled generative factors, enabling precise evaluation.

---

**Algorithm 1** Greedy Dataset Construction via Marginal Contribution

---

**Require:** features $\mathbf{Z} \in \mathbb{R}^{N \times D}$ (from OF-SQAE), full set $\mathcal{X}$, target size $n$, scales $\mathcal{K}$
**Ensure:** subset $\mathcal{X}_S$
 1: $\mathcal{X}_S \leftarrow \varnothing$, $\mathcal{X}_U \leftarrow \mathcal{X}$
 2: Initialize counts $C_{d,k}[i] \leftarrow 0$ for all $d \in [D]$, $k \in \mathcal{K}$, and bins $i$
 3: **for** $t = 1$ **to** $n$ **do**
 4:     $x^\star \leftarrow \varnothing$, $S_{\max} \leftarrow -\infty$
 5:     **for all** $x_u \in \mathcal{X}_U$ **do**
 6:         $S \leftarrow 0$
 7:         **for** $d = 1$ **to** $D$ **do**
 8:             $s \leftarrow 0$
 9:             **for** $k \in \mathcal{K}$ **do**
10:                 compute bin index $b_k$ of $z_{u,d}$
11:                 $s \leftarrow s + \dfrac{1}{C_{d,k}[b_k] + 1}$
12:             **end for**
13:             $S \leftarrow S + s/|\mathcal{K}|$
14:         **end for**
15:         $S \leftarrow S/D$
16:         **if** $S > S_{\max}$ **then**
17:             $S_{\max} \leftarrow S$, $x^\star \leftarrow x_u$
18:         **end if**
19:     **end for**
20:     $\mathcal{X}_S \leftarrow \mathcal{X}_S \cup \{x^\star\}$;    $\mathcal{X}_U \leftarrow \mathcal{X}_U \setminus \{x^\star\}$
21:     update counts $C$ with features of $x^\star$
22: **end for**
23: **return** $\mathcal{X}_S$

---

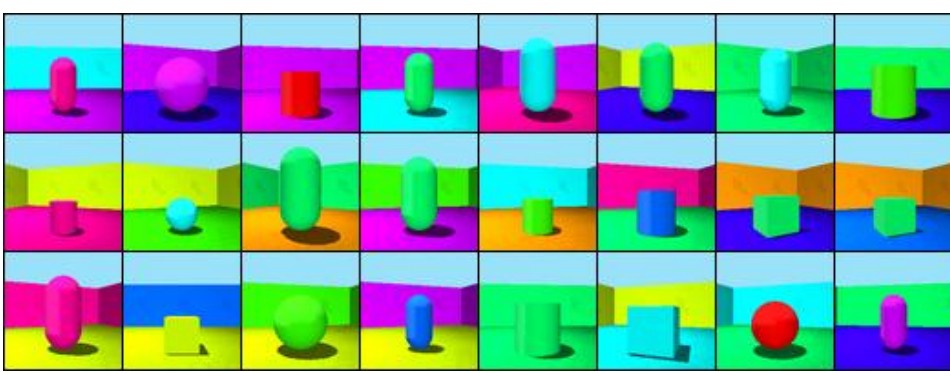

Figure 8: Random samples from the Shapes3D dataset.

**Shapes3D**  Figure 8 shows random samples from the Shapes3D dataset. The dataset contains 480,000 RGB images where six independent factors are varied: floor color (10 hues), wall color (10 hues), object color (10 hues), object size (8 scales), object shape (4 categories), and orientation (15 angles).

**Cars3D**  Figure 9 shows random samples from the Cars3D dataset. The dataset contains 17,568 RGB images rendered from 183 CAD car models. Variations are defined by three factors: car model, camera elevation (4 levels), and azimuth (24 views at $15°$ intervals).

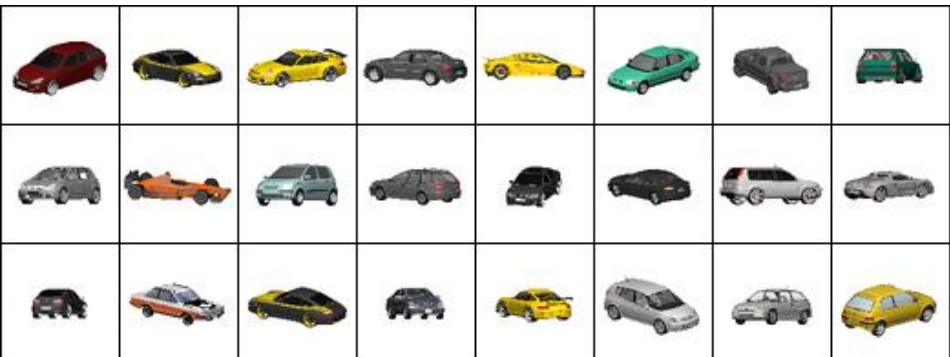

Figure 9: Samples from the Cars3D dataset.

### A.3.2 NATURAL IMAGE DATASETS

These datasets facilitate a comprehensive evaluation of model robustness and generalization.

**STL-10** Figure 10 shows random samples from the STL-10 dataset. The dataset contains $96 \times 96$ color images from 10 object categories, with 5,000 labeled training, 8,000 test, and 100,000 unlabeled samples for unsupervised learning.

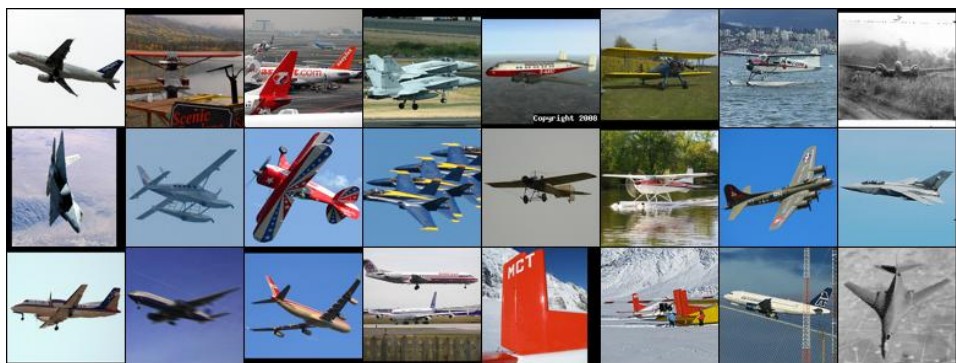

Figure 10: Samples from the STL-10 dataset.

**CIFAR-10** Figure 11 shows random samples from the CIFAR-10 dataset. The dataset contains 60,000 $32 \times 32$ color images across 10 categories, split into 50,000 training and 10,000 test samples.

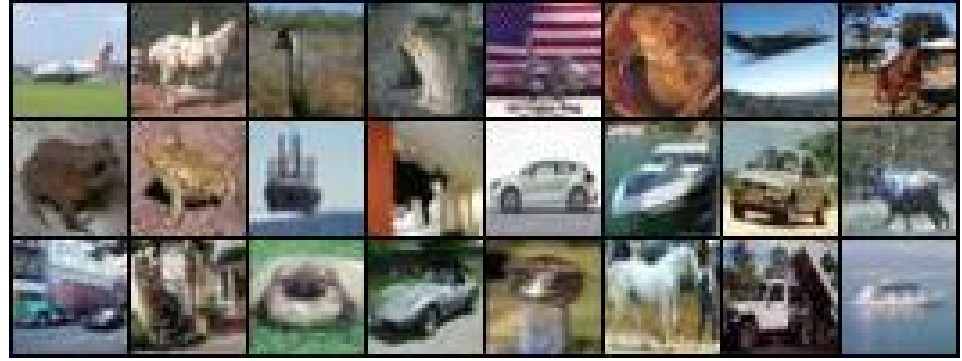

Figure 11: Samples from the CIFAR-10 dataset.

**Imagenette** Figure 12 shows random samples from the Imagenette dataset. The dataset is a 10-class subset of ImageNet with 13,394 images, which we resize to $128\times128$. It provides a challenging yet manageable benchmark for real-world images.

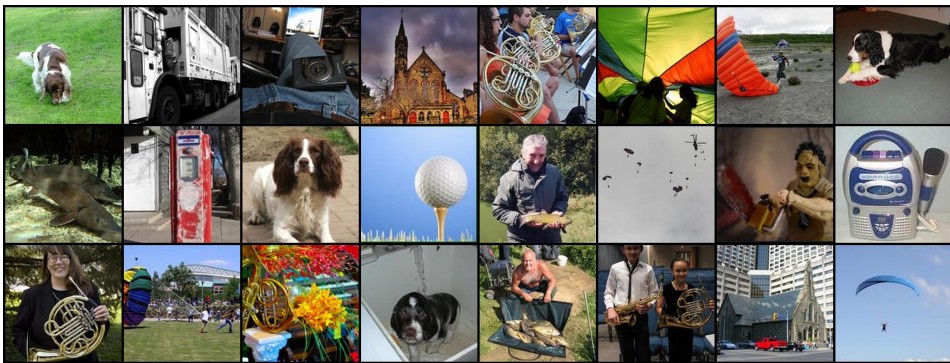

Figure 12: Samples from the Imagenette dataset.

## A.4 Additional Metric Definitions

### A.4.1 IAIQ Validation Metrics

Let $T$ denote the number of semantic factors (tasks) and $N$ the number of validation samples. For factor $t \in \{1, \dots, T\}$ with ground-truth labels $\{y_{n,t}\}_{n=1}^N$ and predictions $\{\hat{y}_{n,t}\}_{n=1}^N$, the (micro) top-1 accuracy is

$$\mathrm{Acc}_t \;=\; \frac{1}{N} \sum_{n=1}^{N} \mathbf{1}\{\hat{y}_{n,t} = y_{n,t}\}. \tag{10}$$

We report the mean accuracy across factors

$$\mathrm{Acc}_{\mathrm{MT}} \;=\; \frac{1}{T} \sum_{t=1}^{T} \mathrm{Acc}_t, \tag{11}$$

and the single-factor case corresponds to $T=1$ (then $\mathrm{Acc}_{\mathrm{MT}} = \mathrm{Acc}_1$). Unless otherwise stated, all accuracies are computed on the full validation set without class reweighting.

### A.4.2 Dataset Construction Generalization Metric

To quantify the *backbone-agnostic* utility of a selected subset, we train $M$ classification backbones on the same subset. When multiple random seeds are used, let $R$ be the number of seeds and $\mathrm{Acc}^{(m,r)}$ the test accuracy of backbone $m$ under seed $r$. Per-backbone accuracy is first averaged over seeds:

$$\overline{\mathrm{Acc}}^{(m)} \;=\; \frac{1}{R} \sum_{r=1}^{R} \mathrm{Acc}^{(m,r)}, \qquad m = 1, \dots, M. \tag{12}$$

We then report the cross-backbone mean and standard deviation:

$$\mathrm{Acc}_{\mathrm{gen}}^{\mathrm{mean}} \;=\; \frac{1}{M} \sum_{m=1}^{M} \overline{\mathrm{Acc}}^{(m)}, \qquad \mathrm{Acc}_{\mathrm{gen}}^{\mathrm{std}} \;=\; \sqrt{\frac{1}{M} \sum_{m=1}^{M} \left( \overline{\mathrm{Acc}}^{(m)} - \mathrm{Acc}_{\mathrm{gen}}^{\mathrm{mean}} \right)^2}. \tag{13}$$

In tables/figures we display results as "$\mathrm{Acc}_{\mathrm{gen}}^{\mathrm{mean}} \pm \mathrm{Acc}_{\mathrm{gen}}^{\mathrm{std}}$".

## A.5 Implementation Details

All experiments were run on NVIDIA L20 GPU servers with Ubuntu 22.04, CUDA 12.4, PyTorch 2.4.1, and Python 3.8. We report task-specific settings for disentanglement and dataset construction below.

### A.5.1 Disentanglement tasks

**Training Paradigm**  The OF-SQAE model is trained in a fully end-to-end manner, where all components are optimized jointly. This unitary training process is applied consistently across all datasets before task-specific hyperparameter tuning.

**Shapes3D**  We use a batch size of 50, a learning rate of $3 \times 10^{-4}$, AdamW , 300 epochs, 10 latent factors, and a codebook size of 20 per latent dimension.

**Cars3D**  We use a batch size of 36, the same learning rate and optimizer, 2000 epochs, 10 latent factors, and a codebook size of 40 per latent dimension.

**Baselines**  For other DRL baselines, we adopt authors official implementations and hyperparameters when available; otherwise, we report the metrics from the original papers.

### A.5.2 IAIQ validation tasks

We verify single-attribute IAIQ on Shapes3D, which contains six semantic factors with class counts [10, 10, 10, 8, 4, 15] (floor color, wall color, object color, object size, object shape, azimuth).

The model is MTAN Net with six task heads. Training uses Adam with a learning rate of $5 \times 10^{-3}$ and a weight decay of $10^{-4}$. We use a batch size of 64 and cross-entropy loss with label smoothing of 0.1. Task losses are averaged. We train for 600 epochs.

### A.5.3 Dataset construction tasks

This section outlines the experimental settings for constructing datasets using our proposed OF-SQAE, followed by the protocol for supervised training and evaluation.

**Unsupervised Training with OF-SQAE**  For the initial unsupervised feature learning phase, we employ the Stochastic Gradient Descent (SGD) optimizer with an initial learning rate of 0.01. The learning rate follows a multi-step schedule, where it is decayed by a factor of 0.5 at epochs 30, 60, and 80. The model-specific hyperparameters for each dataset are as follows:

- **On STL-10:** The model is trained for 1000 epochs with a batch size of 50, 60 latent factors, and a codebook size of 200.

- **On CIFAR-10:** We use the same batch size and latent factor count, but set the codebook size to 100 and train for 1000 epochs.

- **On Imagenette:** The model is trained for 1000 epochs with a batch size of 24, 160 latent factors, and a codebook size of 200.

**Supervised Training and Augmentation**  For the subsequent supervised training phase, we apply standard data augmentations, including random cropping, random horizontal flipping, color jittering, and random grayscale transformation. The training configurations are:

- **On CIFAR-10:** We run for 100 epochs with a batch size of 256.

- **On STL-10:** We run for 200 epochs with a batch size of 64.

- **On Imagenette:** We run for 100 epochs with a batch size of 32.

**Active Learning and Evaluation Protocol**  To simulate a real-world annotation scenario, all methods operate under a fixed budget. They are initialized from the same 5% labeled seed set and proceed for 7 rounds, with an additional +5% of the data labeled in each round. Our DGC method constructs high-information subsets as described in Section III-C, whereas baseline methods employ a ResNet-18 as the feature extractor.

**Generalization Assessment** To rigorously assess the generalization capability of the datasets constructed by each method, we perform a comprehensive evaluation. We additionally train a suite of diverse architectures, including SENet-18 , VGG-11 , Inception , and MobileNet , under identical settings. Each experiment is repeated twice independently, and we report the mean performance to ensure robustness of our results.

