# OpenReview forum: "Disentangled Information Quantification for Dataset Construction in Data-Centric AI"
_ICLR.cc/2026/Conference — ICLR 2026 Conference Withdrawn Submission_

### Official Review · Reviewer_ij42 · 2025-10-24

**Soundness:** 2
**Presentation:** 2
**Contribution:** 3
**Rating:** 2
**Confidence:** 2

**Summary:**

This paper proposes Disentangled Generalizable Construction (DGC) as a model-agnostic method for data selection.
DGC consists of two key components:
(1) OF-SQAE, an autoencoder designed to learn a disentangled embedding space, and
(2) IAIQ, a criterion for selecting a data subset based on the embeddings obtained from OF-SQAE.
Since finding the subset that maximizes IAIQ is computationally intractable, the authors employ a greedy approximation method.

Through experiments on the Shapes3D and Cars3D datasets, the authors demonstrate that embeddings obtained via OF-SQAE achieve higher disentanglement scores (e.g., FactorVAE, DCI) compared to existing methods.
They further evaluate DGC on image classification tasks using multiple CNN architectures and show that DGC-based data selection leads to higher accuracy when the target CNN model changes, compared to active learning methods that select data based on a fixed CNN model.

**Strengths:**

* The paper introduces OF-SQAE, a novel autoencoder capable of learning disentangled embeddings.

**Weaknesses:**

* The claim of being *model-agnostic* appears inconsistent, since DGC fundamentally depends on the OF-SQAE model.
* The details of IAIQ are not sufficiently clear, making it difficult to understand how it functions (see Questions below).
* There is no evaluation of IAIQ as a standalone component.
* Experiments are limited to image classification tasks; thus, the effectiveness of DGC in other domains or task types remains unknown.

**Questions:**

* In Eq. (5), the definition of $k_{\mathrm{eff},j}$ is unclear—specifically, what do “bins” and “scale” refer to?

* Regarding Table 3:

  * To confirm whether existing active learning methods are functioning properly, it would be helpful to include evaluation results when the model is fixed to ResNet18 (the same architecture used for data selection). Including these results in the Appendix would be sufficient.
  * The table caption mentions “All methods,” but since DGC shows different results for the 5% setting, it does not appear to include all methods consistently.
  * How was OF-SQAE trained? Was it trained on all unlabeled data from each dataset?
  * Since methods like CoreSet can work with any embedding function, one could apply CoreSet using embeddings from one of the baselines in Table 1 — or alternatively, apply CoreSet to embeddings from OF-SQAE—to assess how effective the IAIQ criterion itself is within the proposed framework.

* Some abbreviations (e.g., DRL, KL, TC) should be explicitly defined.

* References are presented without parentheses, which makes the paper difficult to read.

* Figure 6 is neither referenced in the main text nor explained.

---

### Official Review · Reviewer_AxzS · 2025-10-28

**Soundness:** 2
**Presentation:** 2
**Contribution:** 2
**Rating:** 4
**Confidence:** 3

**Summary:**

This paper proposes Disentangled Generalizable Construction (DGC), a model-agnostic data selection method that maximizes the semantic coverage of dataset subsets. Specifically, the method first encodes each data point into a disentangled representation using the Orthogonal Factorization Soft-Quantization Autoencoder (OF-SQAE), followed by Invariant Attribute Information Quantification (IAIQ), which measures the semantic information content of the data subset. Using a greedy forward selection algorithm, DGC iteratively samples data points that maximize the IAIQ score under a given cardinality limit. The authors demonstrate that this approach consistently outperforms baselines in terms of dataset generalization, while also showcasing the superior disentanglement capabilities of OF-SQAE compared to other methods.

**Strengths:**

- The overall writing is clear and easy to follow.
- Data selection based on semantic coverage instead of specific model performance seems interesting and sound.
- The empirical results demonstrate consistent improvement in data generalization over baselines.

**Weaknesses:**

- The main motivation of this work, stated in the abstract and introduction, is that “prior model-agnostic approaches for data selection are sensitive to initial labeled sets, acquisition batch sizes, and model choices, which hurts stability and reproducibility (L49-50, L94-95).” However, none of the empirical evidence presented supports this claim, and thereby this state is not very convincing. To strengthen the argument, the authors should provide experimental results and an in-depth analysis of the vulnerability of model-based methods and the robustness of their approach.
- Several design choices lack clear justification (e.g., the loss terms in Equation (6), the need for soft quantization, and why disentanglement is necessary). Please refer to the questions below for further clarification.

**Questions:**

**Questions regarding justifications**
- In Equation (6), what are $L_2$ and $L_{perc}$ and why are they necessary? Providing an ablation study to assess their impact would be helpful.
-  In Equation (8), why is quantization required, and specifically, why *soft* quantization instead of discrete quantization? Empirical analysis supporting this choice is required for justification.
- The role of the disentanglement module in this framework is unclear, as IAIQ in Equation (1) can be measured with a monolithic representation. Justification for the need of disentanglement and empirical evidence to support its importance would be crucial.
- While the primary aim of this work is to propose a data selection method that maximizes semantic coverage based on disentangled representation, the ablation studies mainly focus on evaluating disentanglement metrics. How do the performance of the disentanglement modules (i.e., OF-SQAE) affect dataset generalization? For instance, what are the generalization results (as in Table 3) for the configurations shown in Table 2?


**Questions regarding clarification**
- Could the authors clarify L209-210 ($k_{eff,j}$ is the number of non-empty bins in the full dataset at scale $j$ and $c_j$ the count of those bins covered by $\mathcal{X}_S$)? What exactly are bins, and what is meant by scale $j$?
- In Figure 3, what do the red rectangular boxes represent? Also, what do $Z_8$ & $Z_9$ and $Z_4 Z_5 Z_7$ mean? Does this imply that these factors are entangled to represent a single semantic dimension?
- In L407-408, what does it mean by varying the number of azimuth categories from 1 to 15?
- While dataset generalization is examined on natural datasets, disentanglement is evaluated only on synthetic datasets. How well does OF-SQAE perform in terms of disentanglement on these natural datasets?
- It would be more comprehensive if the authors provided an analysis of which samples are actually selected using this method, how they differ from previous methods, and how these differences affect the final performance.


**Minor Fixes**
- In Figure 2, the feature $V \in \mathbb{R}^k$ is a 1-dimensional vector, but it is visualized as 2-dimensional feature.
- (Suggestion) While the primary aim of this paper is to present a model-agnostic data selection method, the order of experimental results (Section 4.2: Disentanglement Evaluation, Section 4.3: Validation of IAIQ, Section 4.4: Dataset Generalization) may cause readers to perceive Section 4.2 as the main result. In my understanding, Section 4.4 presents the core contribution of the paper (the model-agnostic data selection method). I suggest reordering the sections to emphasize the main result.

---

### Official Review · Reviewer_BeTr · 2025-11-01

**Soundness:** 2
**Presentation:** 2
**Contribution:** 2
**Rating:** 4
**Confidence:** 4

**Summary:**

The article addresses the Data-Centric AI (DCAI) problem of constructing high-utility labeled datasets within a fixed annotation budget, ensuring transferability across models and training setups.
Specifically, the authors propose DGC (Disentangled Generalizable Construction), a model-agnostic sample selection framework that attempts to maximize semantic coverage of the data by operating in a disentangled latent space.
The core ideas are: *i)* they introduce IAIQ (Invariant Attribute Information Quantification) to measure per-attribute information of a subset; *ii)* they design OF‑SQAE (Orthogonal Factorization Soft‑Quantization Autoencoder) to learn interpretable axis-aligned latents via orthogonal NMF and per-dimension soft quantization; and *iii)* they use a greedy selector that admits samples with the highest marginal coverage in multi-scale binnings of the latent space.
Experiments on synthetic and natural-image datasets demonstrate that the OF-SQAE model achieves state-of-the-art disentanglement, and the DGC-selected subsets generalize better across multiple downstream classification architectures.

**Strengths:**

The article discusses a concrete limitation of model-dependent active learning, *i.e.*, poor transferability across architectures, and proposes semantic coverage as a stable and model-agnostic criterion for dataset construction.

From a methodological point of view, the three components fit together logically: OF‑SQAE produces disentangled axis-aligned latents; IAIQ quantifies attribute-wise information; and a greedy algorithm uses marginal contributions to maximize coverage under a budget.

OF‑SQAE achieves strong disentanglement on Shapes3D and Cars3D.
On CIFAR-10, STL-10, and Imagenette, the proposed selection often yields the highest accuracy, averaged across backbones, and tends to reduce variance across architectures.

**Weaknesses:**

The authors claim that their method facilitates the construction of "long-term reusable data assets" (line 053).
The experiments demonstrate reusability across different architectures for the same classification task.
However, "long-term reusability" implies broader utility, for instance, across different tasks (*e.g.*, classification, detection, segmentation) or evolving model families.
The current experiments, limited to classification, do not fully support this claim.

Important methods, such as [1], [2], [3], are not discussed in both the related works and experiments.

Although cross-architecture evaluation is used, the experimental design for baselines could be strengthened.
The active learning subsets (BADGE, ALMix, etc.) are all constructed using a ResNet-18 model (line 421).
The subsequent evaluation then trains multiple different architectures on these ResNet-18-selected subsets.
This setup demonstrates that DGC generalizes better than a ResNet-18-specific active learning strategy.
However, a stronger comparison would involve showing, for instance, that a DGC subset outperforms a subset selected by an active learning method using a VGG-11 when both are evaluated on a VGG-11.
Probably, the current setup does not fully rule out the possibility that a model-specific active learning method could outperform DGC if the constructing model and the evaluation model are the same.

Furthermore, since the evaluation emphasizes transfer across architectures, baselines that are explicitly model-dependent may be structurally at a disadvantage.
A stronger comparison would allow baselines to rely on a self-supervised encoder or compare to dataset selection (not active learning) baselines that are inherently model-agnostic.

---------------
*References:*

[1] GLISTER: Generalization based Data Subset Selection for Efficient and Robust Learning (AAAI 2021)

[2] Deep Learning on a Data Diet: Finding Important Examples Early in Training (NeurIPS 2021)

[3] Active Learning on a Budget: Opposite Strategies Suit High and Low Budgets (ICML 2022)

**Questions:**

The authors should:
1) Provide additional experiments to support the claim of creating "long-term reusable data assets".
2) Discuss and, if possible, compare against important baselines such as the one referenced in the weaknesses.
3) Show results where active learning subsets are constructed using the same architecture as the evaluation model. This would test whether DGC outperforms a specialized method in a matched scenario.
4) Include baselines that are not structurally disadvantaged by the cross-architecture evaluation setup (*e.g.*, SimCLR, MoCo, DINO), or at least dataset selection methods that are inherently model-agnostic, since the article emphasizes cross-architecture transfer.

---

### Official Review · Reviewer_5taV · 2025-11-02

**Soundness:** 3
**Presentation:** 3
**Contribution:** 3
**Rating:** 4
**Confidence:** 4

**Summary:**

This paper addresses a core challenge in Data-Centric AI (DCAI)-constructing reusable, high-utility datasets independent of model architectures. The authors propose Disentangled Generalizable Construction (DGC), a model-agnostic sample selection framework that maximizes semantic coverage instead of relying on model uncertainty or confidence. Specifically, the DGC has three different modules: IAIQ provides a metric quantifying per-sample information based on disentangled latent attributes; OF-SQAE provides an autoencoder that learns stable, interpretable, and axis-aligned semantic factors via orthogonal NMF and per-dimension soft quantization; and a greedy selector that chooses samples contributing most to latent-space coverage.

**Strengths:**

1. The idea is interesting; the author wants to build a method that can shift dataset construction from model-dependent active learning to a model-agnostic semantic-coverage paradigm, aligning well with DCAI philosophy, which can better help current large models to find high-quality training data.
2. The paper is well written and easy to follow.
3. The author provides extensive experiments to demonstrate the effectiveness of their methods. Compared to previous methods, DGC can use fewer data to achieve better performance.

**Weaknesses:**

1. The current model mainly focuses on small toy datasets, and the number of categories is limited. The author should test the model on a larger dataset, like ImageNet, to see whether the proposed method can handle more complex scenarios. Moreover, the author should compare with more up-to-date methods.
2. Although DGC claims to be model-agnostic, Fig. 5 still couples the selector and the evaluator within the same backbone (ResNet-18). While this is acceptable as a controlled test of sampling efficiency, it slightly weakens the claim of model-independence, since the disentangled representation (OF-SQAE) is itself model-based. In order to provide a stronger validation, the author should use a different encoder for selection or employ a pre-trained representation space like CLIP.
3. I feel confused about the gap between Fig.5 and Tab. 3. It looks like only using 8% of the dataset in Fig.5  can achieve better performance than 15% of the dataset in Tab. 3. I noticed that the std is large in this scale; the author should report the accuracy for all test models. Moreover, it is better to see whether using VGG11 as a selector yields performance similar to that of ResNet18 when the testing model is VGG11.

**Questions:**

The author should provide more evidence to prove that the selection is model-agnostic.

---

### Note · Authors · 2026-07-17

I have read and agree with the venue's withdrawal policy on behalf of myself and my co-authors.

---

### Meta-Review · Area_Chair_5N47 · 2025-12-19

**Summary:**

Three main concerns were shared across reviewers:
1. Insufficient dataset and task: The paper mainly conducted experiments on classification tasks, while experiments on more realistic, large-scale datasets and more diverse tasks are expected to further validate the effectiveness of the proposed method.
2. The claim of model-agnostic: The paper claims that the proposed method is model-agnostic. After carefully reading the paper, better cross-architecture generalization might be a more proper frame. The model-agnostic claim is problematic because it is unclear whether such generalization is achieved due to a better architecture (the proposed encoder). If the performance is dependent on the specific encoder, it cannot be framed as model-agnostic.
3. Baselines: The compared baselines are active learning methods that depend on specific model architectures. There is a lack of comparison with other methods that do not use the test model architecture when constructing datasets. Furthermore, when the authors compare different methods on disentanglement. Can those encoders be used for selecting data samples?

Besides, there are also concerns regarding the lack of details on the OF-SQAE model and training. Based on the above concerns, I **cannot** recommend acceptance for this version.

**Reviewer Concerns:**

There wasn't a rebuttal.

**Reviewer Scores:**

There wasn't a rebuttal.

---

### Decision · Program_Chairs · 2026-01-26

Reject